# Mapping of QTLs and Screening Candidate Genes Associated with the Ability of Sugarcane Tillering and Ratooning

**DOI:** 10.3390/ijms24032793

**Published:** 2023-02-01

**Authors:** Ting Wang, Fu Xu, Zhoutao Wang, Qibin Wu, Wei Cheng, Youxiong Que, Liping Xu

**Affiliations:** National Engineering Research Center for Sugarcane, Key Laboratory of Sugarcane Biology and Genetic Breeding, Ministry of Agriculture and Rural Affairs, College of Agriculture, Fujian Agriculture and Forestry University, Fuzhou 350002, China

**Keywords:** sugarcane, tillering, ratooning, QTL mapping, linkage marker

## Abstract

The processes of sugarcane tillering and ratooning, which directly affect the yield of plant cane and ratoon, are of vital importance to the population establishment and the effective stalk number per unit area. In the present study, the phenotypic data of 285 F_1_ progenies from a cross of sugarcane varieties YT93-159 × ROC22 were collected in eight environments, which consisted of plant cane and ratoon cultivated in three different ecological sites. The broad sense heritability (*H^2^*) of the tillering and the ratoon sprouting was 0.64 and 0.63, respectively, indicating that they were middle to middle-high heritable traits, and there is a significantly positive correlation between the two traits. Furthermore, a total of 26 quantitative trait loci (QTLs) related to the tillering ability and 11 QTLs associated with the ratooning ability were mapped on two high-quality genetic maps derived from a 100K SNP chip, and their phenotypic variance explained (PVE) ranged from 4.27–25.70% and 6.20–13.54%, respectively. Among them, four consistent QTLs of *qPCTR-R9*, *qPCTR-Y28*, *qPCTR-Y60*/*qRSR-Y60* and *PCTR-Y8-1*/*qRSR-Y8* were mapped in two environments, of which, *qPCTR-Y8-1/qRSR-Y8* had the PVEs of 11.90% in the plant cane and 7.88% in the ratoon. Furthermore, a total of 25 candidate genes were identified in the interval of the above four consistent QTLs and four major QTLs of *qPCTR-Y8-1, qPCTR-Y8-2, qRSR-R51 and qRSR-Y43-2*, with the PVEs from 11.73–25.70%. All these genes were associated with tillering, including eight transcription factors (TFs), while 15 of them were associated with ratooning, of which there were five TFs. These QTLs and genes can provide a scientific reference for genetic improvement of tillering and ratooning traits in sugarcane.

## 1. Introduction

Sugarcane (*Saccharum* spp. hybrids) is widely grown in more than a hundred countries and regions and plays an important role in sugar production. Tillering and ratooning are important agronomic traits for sugarcane genetic improvement, due to sugarcane being a typically ratoon cultivated crop, and ratoon cultivation is thus adopted as a common cropping system in each producing region. Compared to the plant cane, the ratoon cane can save costs at the planting stage and in the process of cultivation management. Additionally, due to the strong root system, the ratoon cane grows faster than the plant cane, resulting in more efficient use of light energy and earlier maturation [1]. Hence, the sugar content in the ratoon cane is mostly higher than that in the plant cane, especially at the early stage of sugarcane maturity. Sugarcane is a renewable crop, and there is a close relationship between tillering and ratooning (Figure 1). Tillering directly affects the compensation for the lack of plants due to the germination of buds both in the plant and ratoon cane, and thus influences the establishment of sugarcane populations and ultimately the number of millable stalks. In addition, the tiller rate in newly planted cane is the most direct indicator of the tillering ability and one of the important indicators for evaluating the ratooning ability in sugarcane [1]. The higher the tillering rate, the better the rooting ability [2]. The ratoon sprouting rate is an important indicator for evaluating the ratooning ability, which directly affects the plant number of the ratoon cane and ultimately the cane yield. Although cultivation management and the environment conditions can affect their phenotypic expression, tillering rate and ratoon sprouting rate are heritable, genetically controlled, and the genetic basis plays a crucial role in sugarcane [1,3]. Therefore, mapping these two traits and exploring the associated genes will help to improve the tillering and ratooning ability of sugarcane varieties, and provide a basis for further research on their genetic structures.

Tillering, an important trait and a unique form of branching in Gramineae plants, has attracted a lot of attention. In rice (*Oryza sativa*), Ren et al. conducted a genome-wide association study (GWAS) on the trait of effective tiller number using a panel of 490 accessions, and 38 quantitative trait loci (QTLs) associated with tillering were identified [4]. In addition, five QTLs associated with tillering were identified by a high-density map in rice recombinant inbred lines (RILs) [5]. In sorghum (*Sorghum bicolor*), one key regulator of tiller elongation named dormancy-associated protein 1 (DRM1) was identified by combining QTL analysis with transcriptome data [6], and Upadhyaya et al. demonstrated that two simple sequence repeat (SSR) markers were associated with the number of sorghum tillers in 242 endemic varieties [7]. In bread wheat (*Triticum aestivum*), Liu et al. identified seven tillering-related QTLs by means of a genetic map of the 2SY (20828/SY95-71) recombinant inbred line population constructed by combined with a wheat 55K array, SSR marker and kompetitive allele-specific PCR (KASP) marker [8]. Although many linkage markers associated with tillering have been reported in Gramineae plants, the works are mainly concentrated on rice, sorghum and wheat. In sugarcane, only one report was directly related to tillering, where 20 SSR markers were used to investigate the linkage for tillers per plant using 28 sugarcane genotypes, and two markers CEMB-14A and CEMB-14B were confirmed to be tightly linked (*p* < 0.01) to tillers per plant [9]. Additionally, an early study reported two QTLs related to the effective stem number, an agronomic trait positively associated with tillering [10].

Sugarcane ratooning is an important agronomic trait mostly due to the reason that the production cost of the ratoon cane is much lower than that of the plant cane. This ability can be affected by several factors, including genotype, soil fertility, cultivation practices, the climatic conditions, especially temperature and raining after harvesting and before the emergence of ratoon, and even the climate at the time of harvest, but genotype plays a crucial role [1]. In terms of sugarcane hybrids, the genetic composition plays the most important role; for example, the content level of *S. spontaneum* kinship obviously affect ratooning ability in hybrids. In addition, after harvesting, the hormone level in the underground part can also influence the emerging of plantlets from the ratoon cane [11]. Due to the complexity of factors affecting sugarcane ratooning ability, there are only few studies on ratooning, especially no report on mapping QTLs related to ratooning.

With the rapid development of sequencing technology and decreasing costs, the third generation of SNP-based molecular markers has been rapidly developed. Compared to the first and second generations, SNP have the advantages of abundant polymorphisms, high coverage density, high genetic stability, co-dominance, and high throughput. In recent years, many SNP chips from different crops have been developed, such as wheat [12], maize (*Zea mays*) [13] and rice [14]. In sugarcane, most of the trait-associated mapping is related to disease resistance. For example, You et al. identified 23 QTLs associated with the resistance of ratoon stunting disease (RSD) in sugarcane through a selfing population [15]. Six QTLs related to sugarcane leaf blight resistance [16] and 31 QTLs associated with sugarcane chlorophyll content [17] were identified based on the genetic map derived from the Axiom Sugarcane 100K SNP chip.

In the present study, the phenotypic data of the tillering and the ratoon sprouting were investigated in the parents and their 285 F_1_ progenies, derived from a cross of sugarcane varieties YT93-159 × ROC22, in the plant cane and the first and second ratoon cane, cultivated in three different ecological sites with total eight environments. Furthermore, mapping of QTLs and screening candidate genes associated with tillering and ratooning ability was carried out based on the high-quality genetic maps constructed from this population using the 100K SNP microarray by our research group [16].

## 2. Results

### 2.1. Phenotypic Heritability and Correlation Evaluation

Based on the phenotypic data of F_1_ populations and two parents from eight environments, the results of an independent sample t-test revealed a significant difference between parents YT93-159 and ROC22 in the tillering rate of the plant cane, but the difference in the sprouting rate in the first and second ratoon cane was not significant (Table 1). Then, descriptive statistical analysis of phenotypic data was carried out, and it was concluded that there was continuous variation in the frequency distribution of data in each environment (Figure 2). The Kolmogorov–Smirnow test for phenotypic data collected in eight environments revealed that the *p* values were all greater than 0.05, indicating that the distributions of the tillering rate and sprouting rate in the first and second ratoon conformed to the normal rules (Table 1, Figure 3). The tillering rate and the ratoon sprouting rate of the individuals in the F_1_ population varied over a wide range in every crop, and there was an obvious bidirectional super-parental separation. The broad sense heritability (*H^2^*) of the tillering and the ratoon sprouting was 0.64 and 0.63, respectively, indicating that they were middle to middle-high heritable. The positive correlation of the tillering rate among three ecological sites was significant, and so was the first ratoon sprouting rate, suggesting that the genetic variation of the two traits were relatively stable. However, the correlation of the second ratoon sprouting rate between Dehong and Baise was not significant, suggesting that other factors affecting ratoon sprouting may exist during the cultivation of ratooning. Additionally, a significantly (*p* < 0.001) positive correlation between each two of the parameters, i.e., the tillering rate, the first ratoon sprouting rate and the second ratoon sprouting rate, was observed in the same ecological site (Figure 4). These suggest that the ratoon sprouting is positively influenced by the tillering of the plant cane, i.e., the higher the tillering rate in the plant cane, the higher the ratoon sprouting rate in the ratoon cane.

### 2.2. Mapping QTLs Associated with Sugarcane Tillering and Ratooning

Based on two high-quality genetic linkage maps derived from a 100K SNP chip, QTL analysis was performed on the data of tillering, first and second ratoon sprouting collected in eight different environments. Twenty-six QTLs associated with tillering, including two major QTLs and 24 minor QTLs (Table 2), were mapped. Among them, ten were mapped on the male parent ROC22 (Figure 5A), and 16 on the female parent YT93-159 (Figure 5B). The variation range of their logarithm of odds (LOD) values was 2.91–15.68%, and the phenotypic variance explained (PVE) variation was from 4.27 to 25.70%, containing two major QTLs *qPCTR-Y8-1* and *qPCTR-Y8-2*, with the PVE 11.90% and 25.70%, respectively. At the same time, 11 QTLs related to the ratoon sprouting rate, including two major QTLs and nine minor QTLs (Table 3), were mapped, of these, five on the ROC22 (Figure 5A) and six on the YT93-159 (Figure 5B). Additionally, the LOD variation ranged from 3.12 to 5.63%, much smaller than those associated to tillering rate, and the PVE variation range was 6.20–13.54%, including two major QTLs *qRSR-R51* and *qRSR-Y43-2*, with the PVE 13.54% and 11.73%, respectively. It is worth noting that four consistent QTLs were detected, of which *qPCTR-R9*, *qPCTR-Y28* and *qPCTR-Y28* were mapped in the plant cane in two different ecological environments, and another one (*qPCTR-Y60/qRSR-Y60*) was detected in both the plant cane and ratooning environment. In addition, the PVEs of the consistent QTL *qPCTR-Y8-1/qRSR-Y8* were 11.90% in the plant cane and 7.88% in the ratooning environment. These QTLs can be considered as the stable QTL.

### 2.3. GO and KEGG Analysis of Genes

A total of 133 genes were extracted from the four major QTLs of *qPCTR-Y8-1, qPCTR-Y8-2, qRSR-R51 and qRSR-Y43- 2* and four consistent QTLs of *qPCTR-R9, qPCTR-Y28, qPCTR-Y60/qRSR-Y60 and PCTR-Y8-1/qRSR-Y8.* First, 63 genes extracted from QTLs associated with tillering were subjected to GO annotation. These genes played different roles, with cell composition mainly enriched in organelle components and cell part, molecular function mainly in catalytic activity and transport activity, and biological process mainly enriched in the metabolic process and cellular process (Figure 6A). KEGG pathway annotation revealed that tillering associated 43 genes involved in four classes of primary metabolic pathways, which mainly enriched in process of metabolism (Figure 6C). It is interesting that GO annotation of 39 genes associated with ratooning ability had the similar functional results to those for the tillering trait, but played a lesser role in biological functioning and were mainly enriched in metabolic and cellular processes (Figure 6B). In addition, KEGG pathway annotations also yielded similar results to those for tillering, with a total of 30 genes involved in four classes of primary metabolic pathways and mainly enriched in metabolic processes (Figure 6D).

### 2.4. Screening of the Candidate Genes Related to Tillering and Ratooning

A total of 25 genes were preliminarily screened as the candidate genes, including eight transcription factor (TF) genes (Table 4) and 17 protein genes (Table 5), all associated with tillering, though only 15 were associated with the ratoon sprouting rate. These transcription factors belonged to seven classes, including GRAS, C2H2, ERF, C3H, bZIP, Trihelix and NAC families, and among them only four (GRAS, C2H2, CH3 and ERF-like) were associated with ratooning ability. It is thus speculated that these transcription factors may regulate the processes of sugarcane germination and/ or tillering.

In addition, 133 genes were annotated by BLAST in the NCBI database, resulting in 17 proteins related to the target traits tillering or ratooning, all presented in the plant cane but only 10 out of them were identified in the ratoon environments. These 17 proteins belonged to Germin-like, zinc finger CCCH domain-containing, DELLA, Ubiquitin-conjugating enzyme E2, E3 ubiquitin-protein ligase RLIM, Histone-lysine N-methyltransferase ATX3, ATX4, etc. It is inferred that they may play an important role in the growth and development of plants.

## 3. Discussion

Tillering and ratooning, which are affected by a combination of genotypic, cultivated and environmental factors and their interaction effects, are important agronomic traits for sugarcane cultivation. It is difficult to accurately evaluate the above ability of sugarcane varieties due to the complexity and diversity of the environmental factors. Even so, in order to obtain the phenotypic data as accurately as possible, it is essential to carry out multi-site and multi-year field planting experiments. In the preset study, the phenotypic data were collected in eight different environments from three ecological sites distributed in Guangxi, Yunnan, and Guangdong provinces, the three main sugarcane production areas in China, and the data on ratooning ability were collected in the first and the second ratoon cane. As germination will affect the construction of population in the plant cane, and this effect will further affect the ratoon cane, the individuals of 285 F_1_ progenies with a germination rate below 30% in the plant cane were eliminated to increase the reliability of the results of associated mapping. It is assumed that the data are relatively credible.

Selecting plants that meet the breeding objectives is a fundamental part of plant breeding. Currently, sugarcane breeding still relies on a huge, segregated hybrid population; for example, a total of 600,000-1,000,000 seedlings are planted in the field each year for cross breeding in China, since the probability of pyramiding excellent genes is very low, about 1/100,000-300,000, and the selection of superior individuals mainly depends on phenotypic traits in the hybrid population. Compared with traditional phenotypic selection, MAS (marker-assisted selection) can shorten the breeding cycle and improve the accuracy of selection, resulting in a higher breeding efficiency. Especially, if the linkage can be verified in different genetic background, those QTLs identified should have the potential application ability. However, till today only the markers related to the brown rust resistance gene *Bru1* have been successfully used in sugarcane cross breeding [18,19,20], mostly due to the complexity of sugarcane genome.

Compared to diploids and other crops, sugarcane varieties with undeciphered genomes are highly heterozygous, auto-/allo- polyploid and have indeterminate ploidy of 10×–12×, which causes the genetics research to largely lag behind. Therefore, the progress of sugarcane improvement is slow. Fortunately, the traits of sugarcane F_1_ population from crossing is widely separated and can stably inherit in the process of asexual reproduction.

In this study, two high-quality sugarcane genetic maps, which were composed of the female parent YT93-159 covering a length of 4485.2 cM and with 3.0 cM per marker, and the male parent ROC22 covering a length of 2720.0 cM and with 3.5 cM per marker [16], were used for mapping QTLs associated with the agronomic traits of tillering and ratooning. In terms of the population size for QTL localization in sugarcane, the typical size ranges from 100-200 offspring. For example, 151 offspring in total from the cross of SP80-3280 × RB835486 were used for mapping QTL related to four different yield traits, and seven QTLs were obtained [21]. Yang et al. used 171 offspring for mapping QTL related to sugarcane orange rust, and three QTLs were screened [22]. In addition, Aljanabi et al. successfully mapped QTLs for yellow spot disease (*Mycovellosiella koepkei*) resistance in sugarcane by constructing a genetic map using AFLP and SSR markers for 227 F_1_ offspring from the M134/75 × R570 cross [23]. The population of 285 F_1_ progenies used in this study is larger than most previous studies [21,22,23,24,25], but comparative to that of You et al. which contained 296 individuals [26].

In our study, the broad-sense heritability of the tillering and the ratoon sprouting was 0.64 and 0.63, respectively, which is lower than that of sugarcane yellow leaf disease resistance (0.92) [26] and sorghum tiller number (0.78) [7], similar to that of the chlorophyll content (0.66) in sugarcane [17], but higher than that of the number of green leaves (0.43) [27] in sugarcane, and the effective tiller number in rice (0.58) [4] and bread wheat (0.40) [8]. We also confirmed that both tillering and ratooning of sugarcane demonstrated middling to upper-middling heritability, but were not highly heritable traits, indicating that they are not only determined by genetic factors, but also affected by environmental conditions.

In terms of marker number and its precision, here the PVE of the 16 QTLs associated with tillering ranged from 4.27–25.70%, when the LOD was above 2.9, higher than the commonly settled threshold value of 2.5 [26,28,29,30], and with a cumulated 106.99%, which was much lower than the cumulated PVE (195.57%). The number of the 37 QTLs (PVE: 12.31–31.11%) was associated with the tiller number of *Agropyron cristatum* [31], but slightly higher than those identified in sorghum, which ranged from 9.79–17.22% for eight QTLs, with a cumulated 113.09% [6]. However, compared with the trait of tillering in our study, both the cumulated PVE (88.98%) and each PVE (6.20–13.54%) of QTLs (11) associated with ratooning were smaller. When the distance of the nearest marker was taken into account, the distance of the major QTL *qPCTR-Y8-2* (1.5 cM) was much smaller than the average distance of SNP markers in the map of YT93-159 (3.0 cM). However, the distance of the other two major QTLs *qRSR-R51* and *qRSR-Y43-2* associated with ratooning ability was 20.8 cM and 29.8 cM, respectively, which was much greater than the average distance in the map of YT93-159 (3.0 cM). Therefore, the markers associated with tillering are more accurate than that of the ratooning.

Transcription factors play an important role in shoot growth, plant growth and development [32,33]. Here, four classes of transcription factors, GRAS, C2H2, C3H and ERF, were predicted to be associated with tillering and ratooning ability in the consistent QTL *qPCTR-Y8-1/qRCER-Y8*. Among them, C2H2 zinc finger protein can affect root development through the IAA pathway in rice [34]. The ERF family is a major subfamily of the APETALA2/ethylene responsive factor (AP2/ERF) family that can affect growth and development through the ethylene pathway. It is worth mentioning that the candidate gene *Soffic.03G0018200-4E* predicted from the consistent QTL *qPCTR-Y8-1* (*qRSR-Y8*) is a class of GRAS transcription factors. GRAS is widely distributed in plants and plays an important role in rhizome development, meristem formation, and gibberellin signaling [35,36]. Six genes (*LOC_Os11g03110*, *LOC-Os11g04400*, *LOC-Os11g04570*, *LOC-Os05g49930*, *LOC-Os04g49110* and *LOC-Os01g71970*) in the GRAS family were reported to regulate rice tillering [37]. Thus, the *Soffic.03G0018200-4E* gene, belonging to the GRAS family, may have a similar function, such as regulating sugarcane tillering, which needs further evidence.

Exogenous GA signal can affect the process of sugarcane tillering [38], and GA3 treatment stimulates the growth of sugarcane buds, which can improve the germination rate, increase plant height and stem weight, and control the growth of plants [39,40]. In rice, the *OsIAA17q5* gene belonging to GA was identified to control the tiller number [41], and the synthesis of strigolactone (SL) was regulated by GA signaling, while the specific receptor protein of SL could bind to the DELLA protein to form a complex that indirectly regulates rice tillering [42]. It is reasonable to deduce that DELLA protein DWARF8 (*Soffic.03G0038370-1E*) detected in our study may play a similar function in sugarcane tillering. The candidate genes associated with tillering encode proteins, including the Transducin/WD40 repeat-like superfamily protein (*Soffic.03G0018460-3E*), ubiquitin-binding enzyme E2 (*Soffic.03G0018620-2E*) and ubiquitin-linked enzyme E3 (*Soffic.03G0019180-3E*). Among them, the Transducin/WD40 interacts with ribosomal-biogenesis proteins to control seed germination in *Arabidopsis* [43], while the protein ubiquitination plays an important role in several stages of plant development, such as seed dormancy and germination, and root growth [44]. These candidate genes can be used as the priority genes for further research.

## 4. Materials and Methods

### 4.1. Plant Material

A total of 285 F_1_ progenies from the cross (YT93-159 × ROC22) and their parents were included in the phenotypic investigation. They were planted in Dehong, Yunnan (24.43° N, 98.58° E) and Baise, Guangxi (23.75° N, 106.9° E) in March 2020, and Zhanjiang, Guangdong, China (21.27° N, 110.35° E) in March 2021. At the same ecological site, considering the comparability of the agronomic traits between individuals in this population, the best selection is planting them in the same field/plot, which can help to reduce the differences between individuals under test. Due to the differences in the plot size among the three ecological experimental sites, the plot size in different ecological sites was different. Additionally, the row spacing and the seeding density were set according to the planting habits of local sugarcane production. Therefore, the Dehong study adopted a three-repetition random block design, a single row of 1.0 m in length and 1.0 m in row spacing. In Baise, the field trial was designed by a contrast method, a single row of 3 m in length and 1.2 m in row spacing. In Zhanjiang, a random block design with two replicates was adopted, and 5.0 m in length, 1.2 m in row spacing. The planting density was 14 buds in Dehong, 12 buds in Baise and 10 buds per meter in Zhanjiang, respectively. The phenotypic data of the tillering and the ratoon sprouting in the first and second ratoon cane in three ecological sites, with a total of eight environments, were investigated. The phenotypic data were subsequently used for mapping QTLs related to sugarcane tillering and ratooning ability. Routine field management was adopted.

### 4.2. Phenotypic Investigation and Data Analysis

In the plant cane, germination (the number of basic plantlets) and the total tiller number data were recorded 45 and 90 days after sowing. Tillering rate of the plant cane (%) = (The total tiller number/the number of basic plantlets) × 100% [45]. In the ratoon cane, the number of stools germinated was investigated at 60 days after harvest (DAH) of the last crop round [46]. Ratoon sprouting rate (%) = Number of the stools germinated in the ratoon crop/the number of basic plantlets) × 100%. In addition, those individuals with a germination rate of less than 30% in 285 F_1_ progenies were eliminated. IBM SPSS^®^ V25 software (https://www.ibm.com) (accessed on 15 July 2022) was used to analyze the mean, minimum (Mix), maximum (Max), coefficient of variation (CV) of the F_1_ population and its parents. Kolmogorov–Smirnow test for conformity of the sample to a normal distribution and correlation between environments. Normal distribution and correlation plots were drawn using Origin2022b software (https://www.originlab.com) (accessed on 17 July 2022). Broad-sense heritability (*H*^2^) was calculated using QTL Ici-Mapping V4.2 software [47], and the formula is as follows.
$$H^{2}=\sigma_{g}^{2}/(\sigma_{g}^{2}+\frac{\sigma_{ge}^{2}}{n}+\frac{\sigma_{e}^{2}}{nr})$$
where $\sigma_{g}^{2}$ is the estimation of the genotypic variance; $\sigma_{ge}^{2}$ is the estimation of the variance of genotype × environments interaction effect; $\sigma_{e}^{2}$ is the estimation of error variance (or residual variance); n is the number of environments; and r is the number of repetitions.

### 4.3. Mapping of QTLs Associated with Tillering and Ratooning

Two genetic maps used in the QTL analysis were provided by our research group. A total of 93 and 92 linkage groups were constructed for YT93-159 and ROC22, respectively, with 1497 markers covering 4,485.2 cM in length for YT93-159 and 776 markers covering 2720 cM in length for ROC22 [16]. Combined phenotypic data of the tillering and the ratoon sprouting in the first ratoon cane and the second ratoon cane in three ecological sites, with a total of eight environments, were investigated. The comprehensive interval mapping inclusive composite interval mapping (ICIM) method in the software GACD [48] was used for mapping QTLs, and the mapping parameters were set as 1.0 cM (Scanning for the likelihood of QTL presence at 1 cM intervals on each chromosome), LOD manual input 2.9, and other parameters as default. According to the PVE of the output results, those with the PVE value being greater than or equal to 10% were considered as the major effective QTL, otherwise the minor QTL. QTLs detected repeatedly in two or more environments were considered as consistent. According to a previous report [17], the QTL naming rule was as follows: “*q*” + trait + parental linkage group number, and the font is italic.

### 4.4. Gene Extraction and Functional Annotation

The genes located in the intervals of the major and consistent QTL were mined and annotated, and MobaXterm software (https://mobaxterm.mobatek.net/) (accessed on 25 July 2022) was used to write scripts. The probe sequences of the major- and consistent QTLs were compared with the genome annotation file of the *S. officinarum* LA Purple (https://www.ncbi.nlm.nih.gov/bioproject/744175) (accessed on 25 July 2022). Due to the large sequence interval of some loci comparison, the obtained genes may not be precise and accurate enough. Therefore, in this study, only the ID numbers of all genes in the sequence region with the best quality, the highest consistency, and the sequence size within 3 Mb in the comparison results were extracted. The extracted sequences were subjected to GO and KEGG functional analyses (http://www.biocloud.net/) (accessed on 10 August 2022) and those extracted were annotated by the Plant Transcription Factor Database Plant TFDB5.0 (http://planttfdb.gao-lab.org/) (accessed on 19 August 2022) and the NCBI (https://www.ncbi.nlm.nih.gov) (accessed on 25 August 2022) database protein sequence BLASTp tool.

## 5. Conclusions

In the present study, we aim to elucidate the genetic mechanism and lay the foundation for developing molecular markers associated with sugarcane agronomic traits of tillering and ratooning, whose *H*^2^ were estimated to be 0.64 and 0.63, respectively. Two high-quality genetic maps of YT93-159 and ROC22 were then used to locate 37 QTLs based on the phenotypic data collected in eight environments. Among them, 26 QTLs were associated with tillering with the PVEs of 4.27–25.70%, and 11 QTLs were related to ratooning with the PVEs of 6.20–13.54%. In addition, four major QTLs and four consistent QTLs were detected, among which one major QTL *qPCTR-Y8-2* had the smallest distance of 1.5 cM. Furthermore, 25 candidate genes were identified. It is interesting that all these genes were associated with tillering, including eight TFs and 17 protein genes, while 15 of these 25 genes were also associated with ratooning, including five TFs. These QTLs and genes can be used as an important source for further genetic improvement of tillering and ratooning in sugarcane. 

## Figures and Tables

**Figure 1 ijms-24-02793-f001:**
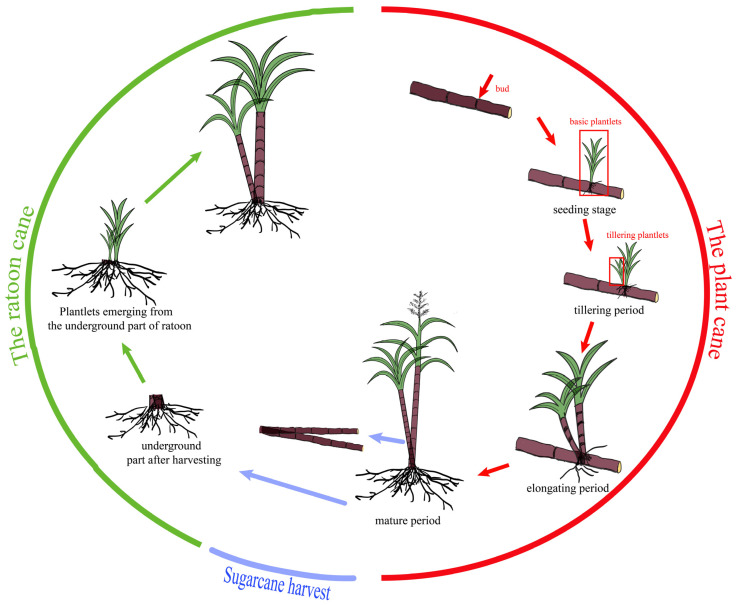
Diagram of sugarcane growth cycle pattern.

**Figure 2 ijms-24-02793-f002:**
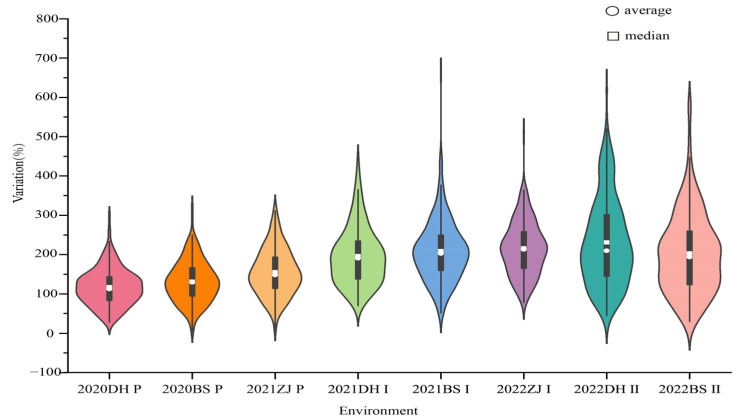
The variation in the tillering rate and the ratoon sprouting rate of 285 F_1_ progenies in different environments. DH, BS, ZJ represent three ecological sites of Dehong, Baise and Zhanjiang province, respectively; *p*, Ⅰ and Ⅱ represent the tillering rate in the plant cane, the first ratoon sprouting rate and the second ratoon sprouting rate, respectively.

**Figure 3 ijms-24-02793-f003:**
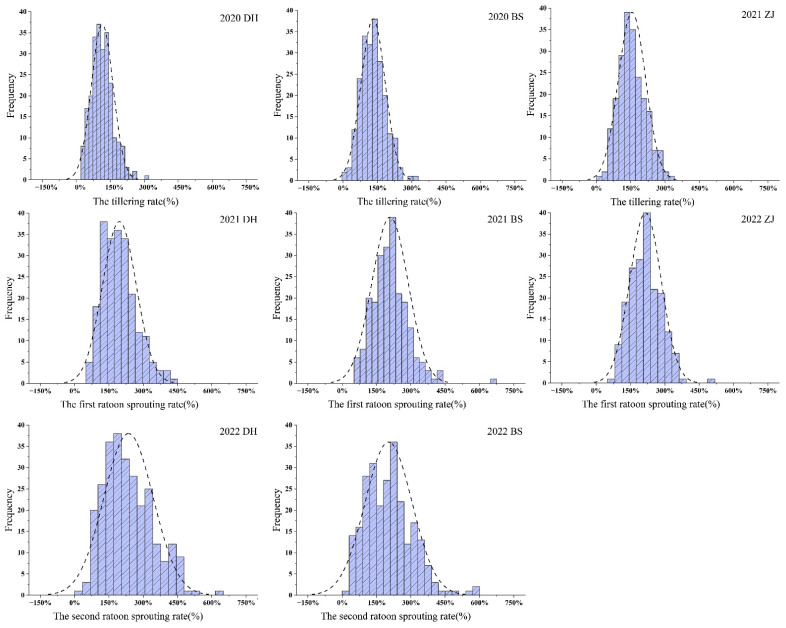
Frequency distribution of the tillering rate and the ratoon sprouting rate of the F_1_ population in the plant cane and the ratoon cane, respectively.

**Figure 4 ijms-24-02793-f004:**
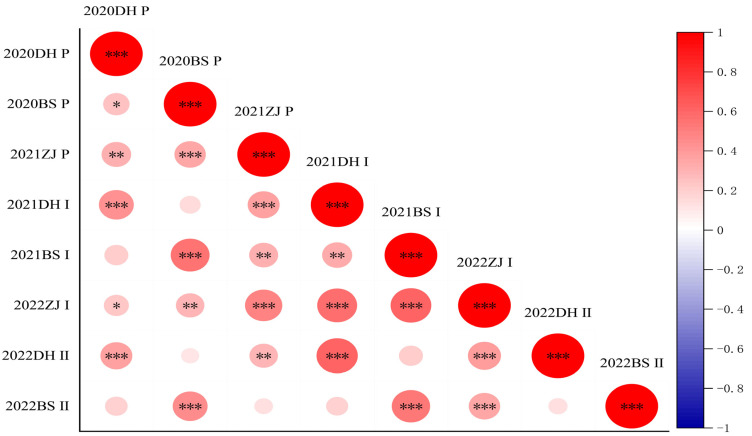
Correlation analysis of the tillering rate and the ratoon sprouting rate of the F_1_ population in the plant cane and ratoon cane under different environments. *, ** and *** indicates significant difference at the 0.05, 0.01 and 0.001 levels, respectively; *p*, Ⅰ and Ⅱ represent the tillering rate in the plant cane, and the ratoon sprouting rate in the first and second ratoon cane, respectively.

**Figure 5 ijms-24-02793-f005:**
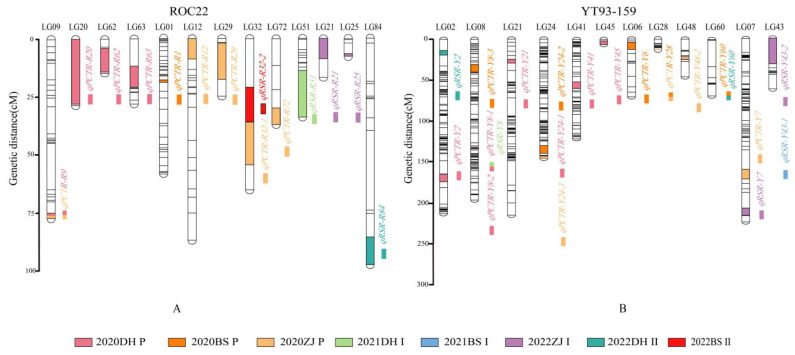
The distribution of the detected QTLs related to the plant crop tillering rate (**A**) and the ratoon sprouting rate (**B**) on sugarcane genetic linkage maps of the parents ROC22 and YT-93-159, respectively. LG, linkage group. Different colors in the figure represent different environments. Pink and orange represent the tillering rate of the plant cane in Dehong and Baise in 2020, respectively; yellow represents the tillering rate of the plant cane in Zhanjiang in 2021; pale green, blue and purple represent the first ratoon sprouting rate in Dehong, Baise and Zhanjiang in 2021, respectively; deep green and red represent the second ratoon sprouting rate in Dehong and Baise in 2022, respectively.

**Figure 6 ijms-24-02793-f006:**
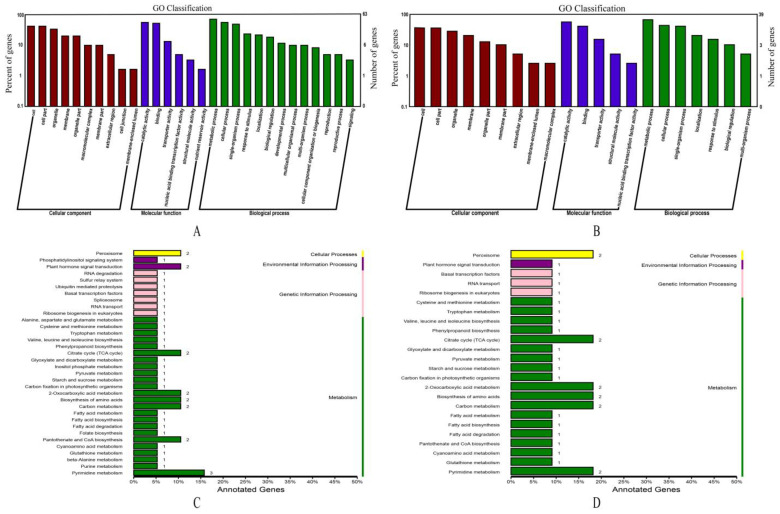
Enrichment analysis of the candidate genes in sugarcane traits of tillering and ratooning associated regions. (**A**,**C**) represent GO functional annotation and KEGG pathway annotation of candidate gene sequences associated with tillering trait, respectively. (**B**,**D**) are similar to (**A**,**C**), but represent sequences associated with ratooning. In the C and D plots, the number after the box bar represents the number of genes annotated in this pathway.

**Table 1 ijms-24-02793-t001:** Statistical analysis of phenotypic data of the parents and the F_1_ population.

Environments		Parents		F_1_ Population				
		YT93-159(%)	ROC22(%)	Max(%)	Min(%)	Mean (%)	CV (%)	K-s (sig)
The plant cane	2020DH	100.0 a	66.7 b	303.3	28.1	117.1	39.9	0.589
	2020BS	104.3 a	58.3 b	325.0	3.3	131.1	40.3	0.656
	2021ZJ	103.5 a	56.3 b	316.7	11.4	155.0	38.5	0.607
The first ratoon cane	2021DH	109.1 a	122.2 a	445.6	69.6	195.8	37.7	0.296
	2021BS	155.6 a	154.2 a	672.7	51.9	208.6	38.1	0.144
	2022ZJ	126.8 a	112.5 a	512.5	78.1	215.5	31.8	0.842
The second ratoon cane	2022DH	118.2 a	144.4 a	620.0	44.4	203.9	48.0	0.118
	2022BS	152.2 a	133.3 a	586.7	29.6	202.5	50.7	0.214

Notes: Different lowercase letters indicate the significant difference between parents at the level of 0.05. DH, BS, ZJ represent three ecological sites of Dehong, Baise and Zhanjiang province, respectively. K-s, Kolmogorov–Smirnow; K-s (sig) indicates all *p* value > 0.05.

**Table 2 ijms-24-02793-t002:** QTLs for the tillering rate of F_1_ population derived from YT93-159 × ROC22 in the plant cane.

Environment	QTL Name	LG	GD (cM)	Left Marker	Right Marker	LOD	PVE (%)	Effect F	Effect M
2020DH P	*qPCTR-R9*	9	2.2	AX-171327125	AX-171310449	3.8035	5.8797	0.1134	−0.0048
	*qPCTR-R20*	20	29.3	AX-171272742	AX-171293480	3.7847	9.498	0.0889	−0.0137
	*qPCTR-R62*	62	11.9	AX-171318447	AX-171292913	3.4595	5.3478	−0.0491	−0.025
	*qPCTR-R63*	63	9.7	AX-171319734	AX-171359264	4.1141	8.1183	−0.0124	−0.0201
	*qPCTR-Y2*	2	9.4	AX-171302044	AX-171244192	3.0729	4.2714	0.0437	0.0813
	*qPCTR-Y8-1*	8	3.7	AX-171329699	AX-171239513	7.7142	11.8968	0.0141	−0.0475
	*qPCTR-Y8-2*	8	1.5	AX-171262083	AX-171274601	15.6779	25.6955	0.0052	−0.0483
	*qPCTR-Y21*	21	4.7	AX-171321699	AX-171270548	3.1302	4.3498	−0.0396	−0.0478
	*qPCTR-Y24-1*	24	1.1	AX-171283687	AX-171367948	3.5537	4.8965	0.0625	0.0559
	*qPCTR-Y41*	41	8.1	AX-171250323	AX-171273872	3.6921	5.4547	0.0409	0.0993
	*qPCTR-Y45*	45	4.3	AX-171306380	AX-171313333	3.4984	4.8351	0.0796	0.0542
2020BS P	*qPCTR-R1*	1	1.1	AX-171351615	AX-171293624	3.0308	6.2062	0.0835	−0.0816
	*qPCTR-Y6*	6	8.7	AX-171257841	AX-171273471	3.1402	5.9345	−0.0309	−0.0951
	*qPCTR-Y8-3*	8	9.4	AX-171324757	AX-171308215	2.9123	5.002	−0.099	0.0519
	*qPCTR-Y24-2*	24	9.4	AX-171283359	AX-171249399	3.558	6.924	0.0231	0.1043
	*qPCTR-Y28*	28	0.7	AX-171326575	AX-171286677	2.914	5.2792	−0.043	0.0097
	*qPCTR-Y60*	60	0.7	AX-171257349	AX-171272905	4.7622	8.8548	0.1576	0.0033
2021ZJ P	*qPCTR-R9*	9	2.2	AX-171327125	AX-171310449	3.0126	5.7327	0.1097	−0.033
	*qPCTR-R12*	12	8.7	AX-171247255	AX-171245486	3.449	7.7469	0.0904	−0.1005
	*qPCTR-R29*	29	17.0	AX-171323413	AX-171256547	3.9984	9.3369	−0.0513	−0.0944
	*qPCTR-R32-1*	32	18.7	AX-171357712	AX-171244747	3.893	7.6342	0.0834	0.1306
	*qPCTR-R72*	72	7.6	AX-171281899	AX-171351113	3.3203	6.2522	0.0937	0.0537
	*qPCTR-Y7*	7	11.8	AX-118065381	AX-171315846	3.6844	6.994	−0.1538	0.0151
	*qPCTR-Y24-3*	24	2.5	AX-171367948	AX-171365860	3.5175	6.0727	0.0961	0.1057
	*qPCTR-Y28*	28	0.7	AX-171326575	AX-171286677	2.9923	5.1918	−0.0946	−0.0355
	*qPCTR-Y48-2*	48	3.2	AX-171357778	AX-171366781	3.3065	6.2547	−0.1493	0.0242

Note: P represents the plant cane; PCTR, plant crop tillering rate; R and Y represent ROC22 and YT93-159, respectively; LG represents linkage group; GD indicates genetic distance between left and right markers; PVE indicates phenotypic variance explained by QTL at the current scanning position; Effect F indicates estimated additive effect of the female parent at the current scanning position; Effect M indicates estimated additive effect of the male parent at the current scanning position.

**Table 3 ijms-24-02793-t003:** QTLs for the ratoon sprouting rate of F_1_ population derived from YT93-159 × ROC22 in the ratoon cane.

Environment	QTL Name	LG	GD (cM)	Left Marker	Right Marker	LOD	PVE (%)	Effect F	Effect M
2021DH Ⅰ	*qRSR-R51*	51	20.8	AX-171263257	AX-117188968	4.3298	13.5387	0.1165	−0.1086
	*qRSR-Y8*	8	3.7	AX-171329699	AX-171239513	3.7314	7.8827	0.1537	0.0250
2021BS Ⅰ	*qRSR-Y43-1*	43	1.4	AX-171352848	AX-171281749	3.1239	6.1969	−0.0471	0.1666
2022Zl Ⅰ	*qRSR-R21*	21	9.1	AX-171350534	AX-171287731	3.1185	7.3514	−0.0233	0.1449
	*qRSR-R25*	25	1.1	AX-171255139	AX-171288321	3.6249	8.5039	−0.1671	0.0768
	*qRSR-Y7*	7	9.3	AX-171313995	AX-171351245	3.8632	8.6699	−0.0769	0.0434
	*qRSR-Y43-2*	43	29.8	AX-171313964	AX-171238129	3.3784	11.7340	−0.0796	0.1491
2022DH Ⅱ	*qRSR-R84*	84	11.5	AX-171312460	AX-171238357	3.2971	7.0035	0.0014	−0.2769
	*qRSR-Y2*	2	5.7	AX-171350044	AX-171252982	3.8197	8.9052	−0.0218	−0.1243
	*qRSR-Y60*	60	0.7	AX-171257349	AX-171272905	3.3165	6.6547	0.2756	−0.0384
2022BS Ⅱ	*qRSR-R32-2*	32	15.3	AX-171261438	AX-171357712	5.6301	9.5406	0.2679	−0.1295

Note: RSR, ratoon sprouting rate; R and Y represent ROC22 and YT93-159, respectively; Ⅰ and Ⅱ represent the ratoon sprouting rate in the first and the second ratoon cane, respectively. LG, linkage group; GD, genetic distance between the left and right markers; PVE, phenotypic variance explained by QTL at the current scanning position; Effect F, the estimated additive effect of the female parent at the current scanning position; Effect M, the estimated additive effect of the male parent at the current scanning position.

**Table 4 ijms-24-02793-t004:** Functional annotation of the transcription factor genes related to sugarcane tillering and ratooning.

QTL Name	Candidate Genes	Gene Description
*qPCTR-Y8-1/(qRSR-Y8)*	*Soffic.03G0018200-4E* ^a^	Gibberellin-ACID insensitive, repressor of gai1-3 and scarecrow (GRAS)
	*Soffic.03G0018550-2P* ^a^	C2H2 zinc-finger protein (C2H2-ZFP) transcription factors (C2H2)
	*Soffic.03G0018550-1P* ^a^	C2H2 zinc-finger protein (C2H2-ZFP) transcription factors (C2H2)
	*Soffic.05G0000090-4P* ^a^	ethylene responsive factor (ERF)
	*Soffic.09G0002040-4E* ^a^	Cysteine3Histidine (C3H) gene family (C3H)
*qPCTR-Y8-2*	*Soffic.02G0010890-1P* ^b^	basic region leucine zipper (bZIP)
	*Soffic.03G0019220-2E* ^b^	Trihelix transcription factors (Trihelix)
	*Soffic.09G0002240-5P* ^b^	NAC transcription factor (NAC)

Note: ^a^ represents genes associated with both tillering and ratooning; ^b^ represents genes associated with tillering.

**Table 5 ijms-24-02793-t005:** Functional annotation of the candidate genes related to sugarcane tillering and ratooning.

QTL Name	Candidate Genes	Gene ID in NCBI	E Value	Gene Description
*qPCTR-Y60/(qRSR-Y60)*	*Soffic.09G0011210-7P* ^a^	XP_025804889.1	0.0	Protein NRT1/ PTR FAMILY 2.3-like [*Panicum hallii*]
*qPCTR-Y8-1/(qRSR-Y8)*	*Soffic.03G0018620-2E* ^a^	NP_001146962.1	7 × 10^−106^	Ubiquitin-conjugating enzyme E2 5A [*Zea mays*]
	*Soffic.08G0000260-3E* ^a^	XP_022678687.1	4 × 10^−43^	zinc finger CCCH domain-containing protein 43-like [*Setaria italica*]
	*Soffic.03G0018530-1P* ^a^	XP_002456072.1	0.0	long chain acyl-CoA synthetase 4 [*Sorghum bicolor*]
	*Soffic.03G0018510-3E* ^a^	XP_008656712.1	0.0	zinc finger CCCH domain-containing protein 13 isoform X2 [*Z. mays*]
	*Soffic.03G0018500-3E* ^a^	NP_001348564.1	0.0	Histone-lysine N-methyltransferase ATX4 [*Z. mays*]
	*Soffic.03G0018500-2T* ^a^	AQK97044.1	0.0	Histone-lysine N-methyltransferase ATX3 [*Z. mays*]
	*Soffic.03G0018460-3E* ^a^	AQK97027.1	0.0	Transducin/WD40 repeat-like superfamily protein [*S. bicolor*]
	*Soffic.03G0018330-3E* ^a^	ONM39830.1	0.0	GDSL esterase/lipase [*Z. mays*]
	*Soffic.03G0038370-1E* ^a^	XP_002458232.2	0.0	DELLA protein DWARF8 [*S. bicolor*]
*qPCTR-Y8-2*	*Soffic.03G0047110-1E* ^b^	XP_002489071.1	5 × 10^−97^	universal stress protein A-like protein [*S. bicolor*]
	*Soffic.09G0002260-3P* ^b^	NP_001334869.1	1 × 10^−102^	ubiquitin-conjugating enzyme E2 36 [*Z. mays*]
	*Soffic.03G0019180-3E* ^b^	PWZ32250.1	0.0	E3 ubiquitin-protein ligase RLIM [*Z. mays*]
	*Soffic.03G0019260-3E* ^b^	XP_002458305.1	0.0	tubby-like F-box protein 1 [*S. bicolor*]
*qPCTR-Y28*	*Soffic.04G0019040-3C* ^b^	XP_002452590.1	0.0	MACPF domain-containing protein At1g14780 [*S. bicolor*]
	*Soffic.04G0018650-2C* ^b^	XP_002454299.1	0.0	RING-H2 finger protein ATL46 [*S. bicolor*]
*qPCTR-R9*	*Soffic.04G0013030-6F* ^b^	XP_002453932.1	9 × 10^-142^	Germin-like protein 2-4 [*S. bicolor*]

Note: ^a^ represents genes associated with both tillering and ratooning; ^b^ represents genes associated with tillering.

## Data Availability

Not applicable.

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
