# Peer review of "Mapping of QTLs and Screening Candidate Genes Associated with the Ability of Sugarcane Tillering and Ratooning"

_ijms, 2023, doi:10.3390/ijms24032793_

Round 1
Reviewer 1 Report
This study presents a QTL analysis of an F1 sugarcane population phenotyped in different environments for traits related to ratooning and tillering ability. Also, an in silico analysis of genes contained within these QTLs was conducted. This yielded multiple candidate genes considered to be involved in or causative for the ratooning or tillering processes.
By itself, the study is of high interest and the design is robust and multiple QTLs/candidate genes have been identified.
However, the objective stated in the text to 'elucidate the genetic mechanism and develop molecular markers for important traits for sugarcane breeding related to tillering and ratooning ability', has not yet been achieved.
While multiple candidate genes have been identified, so far no molecular markers have been developed and a causal relationship between (some of) the QTI’s or candidate genes remains to be formally demonstrated.
Similar QTL approaches and studies including validation of markers for predictive breeding have been published in sugarcane in the area of disease resistance (e.g. orange rust, 2018).
Therefore, adding validated data would be essential to warrant publication. Further, adding validated marker data would strengthen the associations presented through the QTLs and the GO analysis.
Author Response
Dear anonymous reviewer,
We are glad to receive your valuable comments and suggestions to our manuscript. Thank you for your kind consideration on this manuscript "Identification of QTLs and critical genes associated to the ability of sugarcane tillering and ratooning". Without your professional reviews, this manuscript would not be as smooth and more persuasive as what it is now. Thank you very much!
We have amended the manuscript according to all the opinions, suggestions and comments of the reviewers and all the changes have been marked-up in the text by the red fond. The responses to all the comments and suggestions are itemized as follows:
Comment 1: The objective stated in the text to 'elucidate the genetic mechanism and develop molecular markers for important traits for sugarcane breeding related to tillering and ratooning ability', has not yet been achieved. While multiple candidate genes have been identified, so far no molecular markers have been developed and a causal relationship between (some of) the QTI’s or candidate genes remains to be formally demonstrated. Similar QTL approaches and studies including validation of markers for predictive breeding have been published in sugarcane in the area of disease resistance (e.g. orange rust, 2018). Therefore, adding validated data would be essential to warrant publication. Further, adding validated marker data would strengthen the associations presented through the QTLs and the GO analysis.
Revision: Thanks for your valuable comment. As you suggested, I totally agree with the opinion that, if the markers obtained through screening in our research can be clearly verified, the theoretical and practical value of this paper will be greatly improved. However, till now, although many linkage markers associated with tillering have been reported in Gramineae plants, the work are mainly concentrated on rice, sorghum and wheat. In sugarcane, only one report was directly related to tillering, where 20 SSR markers were used to investigate the linkage for tillers per plant using 28 sugarcane genotypes, and two markers CEMB-14A and CEMB-14B were confirmed to be tightly linked (P < 0.01) to tillers per plant. Additionally, an early study reported two QTLs related to the effective stem number, an agronomic trait positively associated with tillering. The commonness of these studies is that the obtained markers have not been well verified for the time being, mostly due to the reason that the genome of sugarcane in autopolyploids is complex, and the incidence of sequence similarity is extremely high. In recent years, third-generation of SNP-based molecular markers has been rapidly developed. Compared to the first and second generations, SNP have the advantages of abundant polymorphisms, high coverage density, high genetic stability, co-dominance, and high-throughput. Recently, many SNP chips from different crops have been developed, such as wheat, maize, rice. In sugarcane, most of the trait-associated mapping is related to disease resistance. For example, You et al. identified 23 QTLs associated with the resistance of ratoon stunting disease (RSD) in sugarcane through selfing population. Six QTLs related to sugarcane leaf blight resistance [16] and 31 QTLs associated with sugarcane chlorophyll content were identified based on the genetic map derived from the Axiom Sugarcane 100K SNP chip. It is no doubt that SNP detection is much more complicated in polyploid such as sugarcane than in diploids, and there is no mature case that can be referred to sugarcane before. Besides, probe based detection is still too costly and not suitable for large sample multi-locus detection now. We are happy to share with you the information that we are currently developing SNP detection methods for homozygous polyploids to ensure the accuracy of detection while keeping the cost under control. What should also be stressed here is that I have a strong resonance that the breeding method and genetic studies of sugarcane cultivars are very different and extremely lagging largely behind other major crops, such as rice and wheat. From this point of view, each minor step to push forward the sugarcane breeding program, if can be encouraged, should be helpful to promote the basic research of sugarcane breeding.
Taken all the above into consideration, one hand, I agree with your opinion that adding validated marker data would strengthen the associations presented through the QTLs and the GO analysis, while on the other hand, we are confident that our existing data is sufficient and effective to support our current results and conclusions. We do hope and believe that in the near future, the validation of these markers obtained in the present study can be clearly validated and then applied into the real process of sugarcane breeding.
As you may know, due to objective reasons that sugarcane is a highly heterozygous allopolyploid plant with a complex genetic background and is lack of genome data, the objective to develop molecular markers for sugarcane breeding related to tillering and ratooning has not yet been achieved due to lack of validation. Therefore, we revised this objective and expressed it as “In the present study, we aim to elucidate the genetic mechanism and lay the foundation for developing molecular markers associated with sugarcane agronomic traits of tillering and ratooning.” in line 412-414. Again we want to express our high and great appreciation for your kind words and professional suggestions, which should no doubt help for our future research. Thanks again.
Any questions, we will be more than happy to answer. Looking forward to hearing from you soon.
Regards and Best wishes!
Youxiong Que and Liping Xu
2022-12-8

Reviewer 2 Report
Comments to authors:
1. In figure 1, is both tillering and ratooning obligated step to complete the lifecycle of sugarcane or either one can be bypass? Is there any difference between sugarcane grew from plant cane or ration cane in term of biomass or sweetness?
2. In figure 2, the variation of the sprouting rate of second ratoon cane seems more significant than in first ratoon cane, it there any reason behind? statistic treatment should be applied here. If possible, please also consider to include 2022ZJ II in the data set.
3. In figure 3, please consider to normalize the scale of both axis for better comparison among different ecological sites and plant stages.
4. In figure 6, the pattern of enrichment of GO and KEGG were high similar between tillering trait (A, C) and ratooning trait (B, D). I wonder wether any differences can be observed like the individual gene in the same GO term or KEGG pathway?
Author Response
Dear anonymous reviewer,
We are glad to receive your valuable comments and suggestions to our manuscript. Thank you for your kind consideration on this manuscript "Identification of QTLs and critical genes associated to the ability of sugarcane tillering and ratooning". Without your professional reviews, this manuscript would not be as smooth and more persuasive as what it is now. Thank you very much!
We have amended the manuscript according to all the opinions, suggestions and comments of the reviewers and all the changes have been marked-up in the text by the red fond. The responses to all the comments and suggestions are itemized as follows:
Comment 1: In figure 1, is both tillering and ratooning obligated step to complete the lifecycle of sugarcane or either one can be bypass? Is there any difference between sugarcane grew from plant cane or ration cane in term of biomass or sweetness?
Revision: Thanks for your valuable comment. Due to the reason that sugarcane is a typically ratoon cultivated crop, it means that both tillering and ratooning are obligated step to complete the lifecycle of sugarcane to guarantee economic benefits. In other words, when performing ratoon planting, each of the steps in Figure 1 is necessary for sugarcane growth. The sweetness of sugarcane is often expressed in terms of Brix. Biomass is correlated with single stem weight, stem thickness, and stem height. From the results of the data in the table, it can be concluded that the F1 population studied in this article does not differ significantly in sweetness, stem height, stem weight and stem thickness under ratoon cane and plant cane. Thanks again.
Table 1 Phenotypic analysis of four yield and quality traits of sugarcane in plant cane and ratoon cane crop seasons
|
Trait |
Plant cane |
|
Ratoon cane |
||||
|
Mean |
SD |
CV (%) |
|
Mean |
SD |
CV (%) |
|
|
Height(cm) |
198.44 |
36.05 |
18.17 |
|
200.85 |
38.34 |
19.09 |
|
Brix(%) |
21.47 |
1.61 |
7.50 |
|
20.62 |
2.17 |
10.52 |
|
Stalk diameter(cm) |
2.55 |
0.31 |
12.16 |
|
2.50 |
0.31 |
12.40 |
|
Stalk weight(kg) |
1.05 |
0.36 |
34.29 |
|
1.01 |
0.35 |
34.65 |
Comment 2: In figure 2, the variation of the sprouting rate of second ratoon cane seems more significant than in first ratoon cane, it there any reason behind? statistic treatment should be applied here. If possible, please also consider to include 2022ZJ II in the data set.
Revision: Thanks for your valuable comment. Because the sprouting rate of second ratoon cane is the plant formed after the first year of harvest, the sprouting rate of second ratoon cane is more easily influenced by environment and has wider phenotypic variation than in first ratoon cane. Due to the one-year growing period of sugarcane, the Zhanjiang planting site was newly planted in 2021. It is not yet possible to investigate data on the sprouting rate of second ratoon cane. However, we are confident that our existing data is sufficient and effective to support our current results and conclusions. Thanks again.
Comment 3: In figure 3, please consider to normalize the scale of both axis for better comparison among different ecological sites and plant stages.
Revision: Thanks for your professional comment. Changes have been made in accordance with your suggestions. All the changes have been marked up in the text by the red fond.
Comment 4: In figure 6, the pattern of enrichment of GO and KEGG were high similar between tillering trait (A, C) and ratooning trait (B, D). I wonder whether any differences can be observed like the individual gene in the same GO term or KEGG pathway?
Revision: Thank you for your comment. In the present study, GO annotation was performed to find out the main functions of the candidate genes. KEGG analysis was then performed to find out which metabolic pathways the candidate genes are involved. A and B, C and D are highly similar, and it is because most of the screened candidate genes are not only associated with tillering but also with ratooning. There are certainly any differences in individual genes within the same annotated classification. For example, two genes (Soffic.09G0002050-2P, Soffic.09G0002100-3P) were annotated in the first metabolic pathway (Citrate cycle (TCA cycle)) of Figure 6(C). Soffic.09G0002050-2P is involved in tricarboxylic acid cycle (GO:0006099); malate metabolic process (GO:0006108); response to cold (GO:0009409) in GO annotation. While Soffic.09G0002100-3P is only involved in the tricarboxylic acid cycle (GO:0006099) and isocitric acid metabolic process (GO:0006102). Not only that, Soffic.09G0002050-2P gene annotation information in NCBI is malate dehydrogenase, while Soffic.09G0002100-3P gene annotation information is NADP-dependent isocitrate dehydrogenase. Therefore, the individual in the same metabolic pathway or the same GO project, and that is why individual genes in the same metabolic pathway or in the same GO project are subject to differences. Thanks again.
Take this opportunity, again we do want to express our high and great appreciation for your kind words and professional suggestions, which should no doubt help for our future research. Thanks again.
Any questions, we will be more than happy to answer. Looking forward to hearing from you soon.
Regards and Best wishes!
Youxiong Que and Liping Xu
2022-12-8

Round 2
Reviewer 1 Report
With regards to the development of molecular markers, the objectives of the study have now been reformulated so that the results presented now better reflect these stated objectives. However, similarly the data presented with regards to the genes identified in the QTL's does not support the conclusions or statements made in the manuscript.
In some parts of the manuscript, it is correctly stated that these are candidate genes. However, in the title and other parts of the these genes are stated as 'critical', suggesting that evidence is presented in the paper for a causal relationship with the traits under study.
Currently, this conclusion is based solely on in silico GO analysis of genes within these QTL's. This analysis returns mostly transcription factors and other (regulatory) genes, some of which could potentially be involved in tillering and ratooning. However, similar (classes) genes are often found when studying other traits such as (a)biotic stress responses, etc. Similar to the marker technologies, stronger evidence is needed to make such claims.
Secondly, numerous QTLs studies have been previously published in other crops that have no relevance for practical breeding or trait genetic studies. Given that the research objectives have been adjusted, the manuscript would greatly benefit by explaining in the discussion why these particular QTL's will prove useful for practical breeding or genetic studies in sugarcane.
Author Response
Dear anonymous reviewer,
Thank you for your kind consideration on this manuscript “Mapping of QTLs and Screening Candidate Genes Associated with the Ability of Sugarcane Tillering and Ratooning” which formerly termed as "Identification of QTLs and critical genes associated to the ability of sugarcane tillering and ratooning". The responses to your comment and suggestion are itemized as follows:
Comment 1: Similarly the data presented with regards to the genes identified in the QTL's does not support the conclusions or statements made in the manuscript. In some parts of the manuscript, it is correctly stated that these are candidate genes. However, in the title and other parts of the these genes are stated as 'critical', suggesting that evidence is presented in the paper for a causal relationship with the traits under study. Currently, this conclusion is based solely on in silico GO analysis of genes within these QTL's. This analysis returns mostly transcription factors and other (regulatory) genes, some of which could potentially be involved in tillering and ratooning. However, similar (classes) genes are often found when studying other traits such as (a)biotic stress responses, etc. Similar to the marker technologies, stronger evidence is needed to make such claims. Secondly, numerous QTLs studies have been previously published in other crops that have no relevance for practical breeding or trait genetic studies. Given that the research objectives have been adjusted, the manuscript would greatly benefit by explaining in the discussion why these particular QTL's will prove useful for practical breeding or genetic studies in sugarcane.
Revision: Thanks for your valuable comment. As you suggested, I totally agree with the opinion that, the genes obtained in our study should be written as how many “candidate genes” were screened, rather than how many “critical” gene. Accordingly, we have also made the corresponding changes in the title and in the full text.
Regarding why do we consider these 25 to be candidate genes, firstly, we extracted 133 genes from the major and consistent QTL. Then, their functional predictions were conducted through the transcription factor database and the NCBI database one by one using blastp, followed by founding them involved in plant tillering, plant growth and development, etc. in the previous published papers. Therefore, we consider these 25 genes to be the candidate genes.
Just as you have seen, GO annotation was performed to find out the main functions of the candidate genes enrichment. KEGG analysis was then performed to find out which metabolic pathways the candidate genes are involved in. In the revised manuscript, in order to present it more clearly, we modified the description from “GO analysis showed that 63 genes were associated with tillering” to “63 genes extracted from QTLs associated with tillering were subjected to GO annotation” in our revised manuscript.
Thanks for your reminding that “Given that the research objectives have been adjusted, the manuscript would greatly benefit by explaining in the discussion why these particular QTL's will prove useful for practical breeding or genetic studies in sugarcane.”. This is to confirm that, about why these particular QTL's will prove useful for practical breeding or genetic studies in sugarcane, firstly, MAS (marker-assisted selection) can shorten the breeding cycle and improve the accuracy of selection, resulting in an improvement of breeding efficiency. And the particular QTLs identified in the present study should have potential application ability in MAS if the linkage can be verified in different genetic backgrounds, thought only the markers related to the brown rust resistance gene Bru1 have been successfully used in sugarcane cross breeding till now (after the related studies having been published in Theoretical and applied genetics in 1996, 2004 and 2012) due to the genome complexity and the undeciphered genome. Anyway, it is the direction of technological innovation in sugarcane cross breeding. Additionally, sugarcane breeding still relies on a huge segregated hybrid population, for example, a total of 600,000-1,000,000 seedlings are planted in the field each year for cross breeding in China, since the probability of pyramiding excellent genes is very low, such as about 1/100,000-300,000, and the selection of superior individuals mainly depends on phenotypic traits in the hybrid population.
Take this opportunity, thanks again for your kind words and useful suggestions, which are helpful to the improvement of our manuscript. Any questions, we will be more than happy to answer.
Looking forward to hearing from you soon.
Regards and Best wishes!
Youxiong Que and Liping Xu
2022-12-16

Reviewer 2 Report
I am satisfied with point-by-point revision what authors reply and do not have further question.
Author Response
Dear anonymous reviewer,
Thank you very much for your valuable comments and suggestions on this manuscript, the quality of manuscript has been greatly improved. This has enabled the manuscript to be published in International Journal of Molecular Sciences. Your professional comments have made our manuscript more convincing.
Regards and Best wishes!
Youxiong Que and Liping Xu
2022-12-16
Round 3
Reviewer 1 Report
as mentioned already in my second review report to the editor(s), if these remarks are addressed, then I leave the final decision for acceptance with the responsible editor(s).
no more comments
Author Response
Dear anonynour reviewer,
Thanks again for your professional review and kind consideration.
This time we again try our best to improve the manuscript.
We do hope that now it fulfills the requirement of acceptance for publication in its present form.
Regards and Best wishes
Youxiong Que and Liping Xu
2023-1-20
